# Ethanol binge drinking exposure affects alveolar bone quality and aggravates bone loss in experimentally-induced periodontitis

Deborah Ribeiro Frazão[1©], Cristiane do Socorro Ferraz Maia[2©], Victória dos Santos Chemelo[1], Deiweson Monteiro[1], Railson de Oliveira Ferreira[1], Leonardo Oliveira Bittencourt[1], Gabriela de Souza Balbinot[3], Fabrício Mezzomo Collares[3], Cassiano Kuchenbecker Rösing[4], Manoela Domingues Martins[5], Rafael Rodrigues Lima[1] *

1 Laboratory of Functional and Structural Biology, Institute of Biological Sciences, Federal University of Pará (UFPA), Belém, Pará, Brazil, 2 Laboratory of Inflammation and Behavior Pharmacology, Pharmacy Faculty, Institute of Health Science, Federal University of Pará (UFPA), Belém, Pará, Brazil, 3 Dental Materials Laboratory, School of Dentistry, Federal University of Rio Grande do Sul (UFRGS), Porto Alegre, Brazil, 4 Department of Periodontology, School of Dentistry, Federal University of Rio Grande do Sul (UFRGS), Porto Alegre, Brazil, 5 Department of Oral Pathology, School of Dentistry, Federal University of Rio Grande do Sul (UFRGS), Porto Alegre, Brazil

© These authors contributed equally to this work.
* rafalima@ufpa.br, rafaelrodrigueslima@hotmail.com

**Data Availability Statement:** All relevant data are within the manuscript and its Supporting Information files.

## Abstract

### Background

Periodontitis is a multifactorial inflammatory disease of tooth supporting tissues caused by oral biofilms, influenced by environmental and genetic factors, among others. Ethanol consumption has been considered a factor that enhances alveolar bone loss, especially in high doses. The present study aims to investigate the changes promoted by ethanol binge drinking per se or associated with ligature-induced periodontal breakdown on alveolar bone loss.

### Materials and methods

Thirty-two Wistar rats were randomly allocated into four groups: control (C), ethanol (3g/kg/day; 3 days On-4 days Off protocol by gavage for 28 days, EtOH), experimental periodontitis (EP) and experimental periodontitis plus ethanol administration (EP+EtOH). On day 14th, periodontitis was induced by ligatures that were placed around the lower first molars. On day 28th, the animals were euthanized and mandibles were submitted to stereomicroscopy for exposed root area analysis and micro-computed tomography (micro-CT) for the evaluation of alveolar bone loss and microstructural parameters.

### Results

The results revealed that ligature-induced alveolar bone loss is aggravated by ethanol binge drinking compared to controls (1.06 ± 0.10 vs 0.77 ± 0.04; p<0.0001). In addition, binge drinking *per se* altered the alveolar bone quality and density demonstrating a reduction in trabecular thickness, trabecular number parameter and bone density percentual.

**Funding:** This study was partially supported by a grant provided by CNPq - Brazilian National Council for Scientific and Technological Development - 435093/2018-5 (DF, VC, RO, DF, LB, RL), CAPES/ PROCAD - Higher Education Improvement Coordination - 23038.005350/2018-78 (CM, GB, FC, CR, MD, RL) and (CAPES) – Finance Code 001. The funders had no role in study design, data collection and analysis, decision to publish, or preparation of the manuscript. The APC was funded by Pró-Reitoria de Pesquisa e Pós-graduação from Federal University of Pará (PROPESP-UFPA). The funders had no role in study design, data collection and analysis, decision to publish, or preparation of the manuscript.

**Competing interests:** The authors have declared that no competing interests exist.

Periodontal disorder plus ethanol binge drinking group also demonstrated reduction of the quality of bone measured by trabecular thickness.

## Conclusions

In conclusion, intense and episodic ethanol intake decreased alveolar bone quality in all microstructural parameters analyzed which may be considered a modifying factor of periodontitis, intensifying the already installed disease.

## Introduction

Periodontitis is a multifactorial inflammatory disease, which affects the tooth supporting tissues (i.e., periodontal ligament, cementum, and alveolar bone). Periodontal destruction has been observed among 60–70% of the global population. As a multifactorial disease, 20% of the clinical findings of periodontitis are related to oral biofilms, however, the other 80% are linked to behavioral, environmental and genetic factors, which can modify periodontitis progression. In addition, immunological host response plays a pivotal role in prognosis of periodontally diseased individuals [1, 2].

Some medical conditions, smoking, and social history are described as disease modifiers. They seem to act altering the expression of periodontal disease. However, the exact role of systemic diseases and exposure to different risk indicators in initiating or modifying the progress of the periodontal disease is complex and not fully understood. Diabetes and smoking are the know true risk factors for periodontal disease. Diabetes is characterized as a progressive factor if the glycemic levels are above ideal [3].

Another factor, classified as behavioral, is smoking, which is associated with increased alveolar bone loss. As smoking is often related to alcohol consumption, the toxicological effects caused by ethanol have been related as possible alveolar bone loss intensified by its consumption [4]. Limited studies have linked alcohol consumption and periodontitis [5–7], however, evidence suggests that the relationship exists with several gaps on the exact interaction between alcohol and periodontitis [5, 8]. In the relationship between alcohol exposure and occurrence and progression of periodontitis, a J-curve has been proposed, in which low doses tend to be associated with lower progression and high doses with a more aggressive pattern of progression. Our group has been working on the association between alcohol intake and oral health consequences in animal models both on a heavy alcohol exposure paradigm [9, 10] and with binge drinking-type protocol [11].

Ethanol harmful effects in the oral cavity seem to be proportional to the alcohol-exposure pattern [7, 12, 13]. Enzymatic induction, immune modifier, as well as cementum formation and osteoblast activity reduction have been suggested to alcohol-related oral cavity health disorders [8, 10].

In addition, binge drinking pattern has been reported as the principal type of ethanol exposure among adolescents and adults [14]. Such "social behavior", characterized by large amounts of alcohol in a short period of time, provokes overspread damages in several organs and tissues, and is not restricted to the central nervous system or liver [15]. The exact pathogenic mechanism that underlies ethanol harmful effects has been extensively investigated, however, it has been postulated that inflammatory and oxidative damage are two important pathways on the alcohol-related tissue damage [16]. Both ethanol pathological via that was described above are shared by periodontitis [17]. The aim of the present study was to assess

the effect of binge drinking on alveolar bone loss in Wistar rats. We hypothesize that pro-longed binge drinking exposure is associated with the development of spontaneous periodontitis. Our second hypothesis relies on ethanol exposure increasing the impairment caused by ligature-induced periodontitis.

## Material and methods

### Animals

Wistar rats (90 days-old; n = 32), obtained from animal facility (Federal University of Pará), were randomly allocated to cages (n = 4), with controlled food (NUVITAL®, 3 pellets/animal) and water *ad libitum*. A 12 h light/dark cycle (lights on 7 AM) and temperature control (25 ±1˚C) was used. All procedures were approved by the Ethics Committee on Experimental Animals of the UFPA (under number 6896071217), according to *NIH Guide for the Care and Use of Laboratory Animals* recommendations.

### Experimental procedures

Fig 1 describes the study protocol. Animals were divided into 4 groups. In the control group (C, G1), animals received distilled water. Ethanol group (EtOH, G2) received ethanol binge drinking protocol characterized by 30% w/v ingestion in 4 sections of 3 consecutive days On-4 days Off protocol by gavage [18]. On the 14th day of experiment, the animals from experimental periodontitis group (EP, G3) and experimental periodontitis + ethanol group (EP + EtOH, G4) were submitted to intraperitoneal anesthesia [xylazine 2% (2mg/ml) plus ketamine 10% (10mg/ml)] and the ligature-induced periodontitis was installed by insertion of cotton ligatures (Coats Corrente, São Paulo, SP, Brazil) around the cervical regions of the first inferior molars, which was maintained until euthanasia, as previously described [19]. Ligatures were maintained for 14 days [19, 20]. Following the ethanol binge drinking protocol, animals were euthanized by intraperitoneal anesthesia [xylazine 2% (2mg/ml) plus ketamine 10% (10mg/ml)] followed by cervical dislocation, and jaws were collected. The right hemimandibules were maintained in physiologic serum (NaCl 0.9%) under the refrigerator for stereomicroscopic analyses. The left hemimandibules were maintained in formol solution 4% for micro-computed tomography (micro-CT) evaluation.

**Stereomicroscopic analyses.**   The right hemimandibles from each animal (n = 8, each group) were examined by a stereomicroscope (Discovery V8 Zeiss, Germany). The samples were immersed in 10% sodium hypochlorite (NaOCl) for three hours and then they were washed in an ultrasonic bath with distilled water for 2 minutes. To make a better differentiation of the cementum-enamel junction (CEJ), the hemimandibles were immersed in 1% methylene blue for 60 seconds. After they dry at room temperature, the samples were fixed with wax n˚ 9 (Lysanda, Vila Prudente, São Paulo, Brazil) on individual glass slides with the lingual face of the teeth perpendicular to the axis of observation. The pictures were obtained with a 6.1-megapixel camera (Cannon, Powershot A640) coupled with the stereomicroscope (30.0X). Then, the measurement of the exposed root area (ERA) was made through these images, captured with an adequate scale, from the lingual face of the first molar.

**Bodyweight evaluation.**   For bodyweight measurement, all animals were weighted weekly.

**Micro-computed tomography (micro-CT) analysis.**   Animals left hemimandibules were submitted to micro-CT (MicroCT.SMX-90 CT; Shimadzu Corp., Kyoto, Japan). Images were captured under a rotation of 360˚, with an intensity of 70kV and 100 mA. After this, images were reconstituted by inspeXio SMX-90CT software (Shimadzu Corp., Kyoto, Japan), with a voxel size of 10 μm and resolution of 1024x1024 which resulted in 541 images per sample. Bone loss was evaluated in height by RadiAnt DICOM Viewer 5.0.1 (Medixant, Poznan,

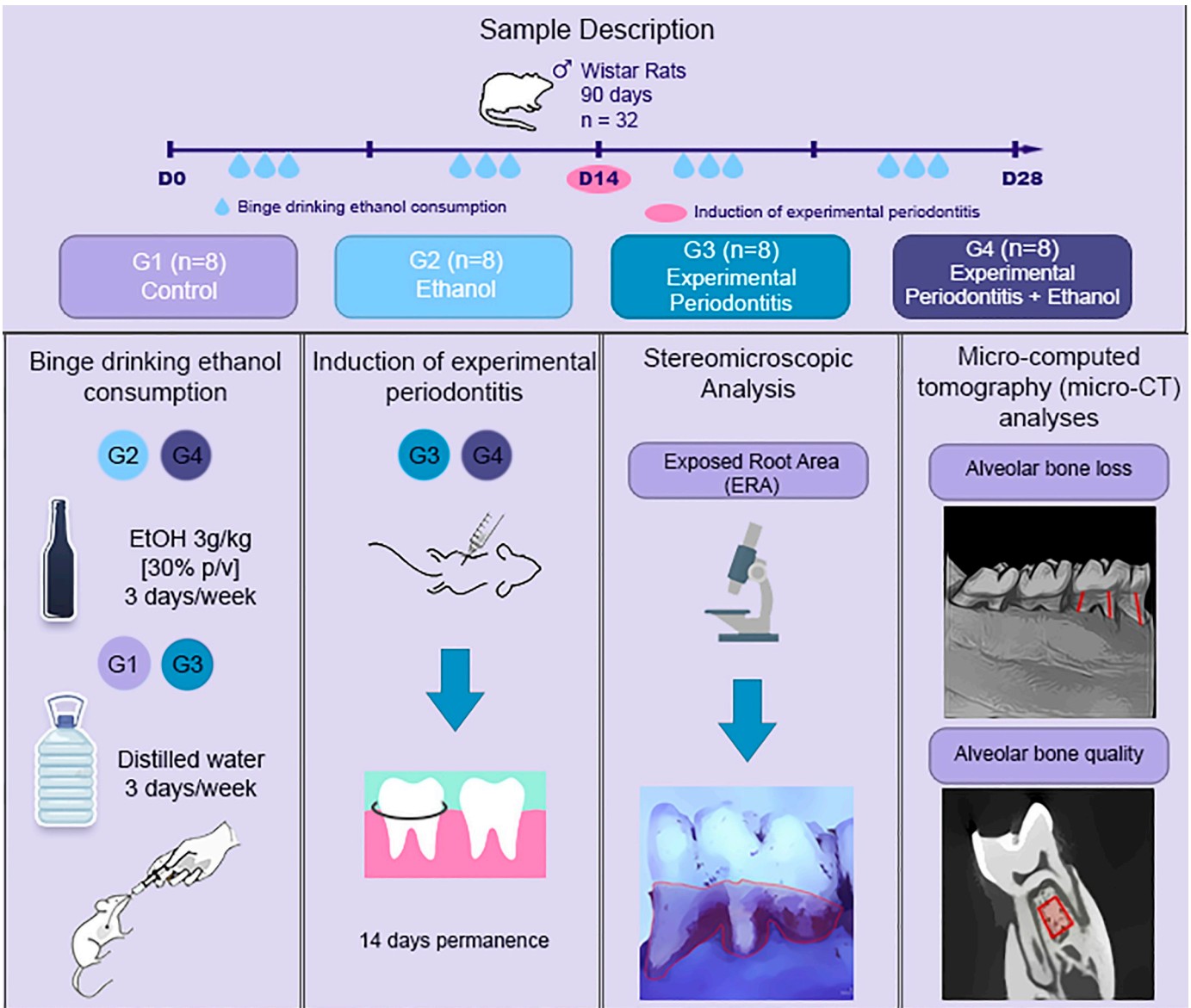

**Fig 1. Sample description and experimental steps.**

Poland) where the 3D reconstruction of the hemimandibulas was performed. The tridimensional models were placed on a standard position (Fig 1), where the vestibular and lingual tooth face could be observed. Thus, the vertical bone loss was detected through the measurement of the distance between the cementum-enamel junction and the alveolar bone crest at six points of the first inferior molar, (i.e., mesio-buccal, buccal, distobuccal, disto-lingual, lingual and mesio-lingual) [21], performing the average of these regions.

To verify the alveolar bone tissue quality, the ImageJ® (National Institutes of Health, Bethesda, MD, USA) software was used on a set of 70 images from the inferior first molar alveolar bone region. The interradicular region, close to the furcation area, from the inferior first molar, including the cervical third until the middle third of the root (average area of $0.160mm^2$), was chosen as the standardized region of interest (Fig 1).

The threshold (0–70) was applied to the segmentation of the different scores of gray color present in the image. The plug-in BoneJ (National Institutes of Health, Bethesda, MD, USA) was used to evaluated the trabecular number (Tb.N), trabecular thickness (Tb.Th), trabecular separation (Tb.Sp), and the percentual of bone density (%BV/TV).

## Statistical analyses

Animals number samples per group were obtained by G*Power software (Statistical Power Analyses 3.1.9.2). Data distribution normality was analyzed by Shapiro-Wilk test. Statistical analyses were performed by one-way ANOVA followed by Tukey test. Bodyweight data was evaluated by repeated measure two-way ANOVA. Student t test was performed to compare two specific groups. All results were expressed as mean ± standard derivation (SD). Statistical differences were adopted when p<0.05. GraphPad Prism 6.0. software (GraphPad, San Diego, CA, USA) was employed for statistical analyses.

## Results

### Bodyweight evaluation

Mean body weight did not reveal statistically significant differences among groups. (control: 246.1 ± 6.29; ethanol: 230.6 ± 3.45; ligature-induced periodontitis: 238 ± 1.98; ligature-induced periodontitis + ethanol: 254.8 ± 3.87; *p* = 0.97).

### Stereomicroscopic analysis

**Exposed root area (ERA).** The stereomicroscopic analyses showed that ethanol binge drinking exposure did not increase the ERA compared to animals that were not exposed to ethanol (2.05 ± 0.08 vs 1.94 ± 0.08; p = 0.90). However, the animals from the EP + EtOH group presented a higher area of exposed root in comparation to ethanol *per se* protocol (2.88 ± 0.21 vs 2.05 ± 0.08; p = 0.0005) and controls (2.88 ± 0.21 vs 1.94 ± 0.08; p = 0.0002). The graphic representation of the stereomicroscopic analysis is shown on Fig 2.

### Micro-computed tomography (micro-CT) analysis

**Alveolar bone loss.** 3D hemimandible analyses show that four ethanol binge drinking exposure did not display additional alveolar bone loss as compared to animals that were not exposed to ethanol protocol (0.82 ± 0.08mm *vs* 0.77 ± 0.04mm; p = 0.49). However, animals submitted to binge drinking protocol plus experimental periodontitis presented an increase in alveolar bone loss compared to ethanol *per se* protocol (0.82 ± 0.08mm *vs* 1.06 ± 0.10mm; p = 0.0001), as well as experimental periodontitis (1.06 ± 0.10mm *vs* 0.92 ± 0.03mm; p = 0.003) and control (1.06 ± 0.10mm *vs* 0.77 ± 0.04mm; p<0.0001) (Fig 3).

**Alveolar bone quality and bone density.** Control group (C) presented the higher trabecular thickness parameter (Tb.Th) values (0.15 ± 0.02mm). Interestingly, four binge drinking exposure in ethanol group showed reduction in Tb.Th even in the absence of experimental-induced periodontitis (p = 0.0468). In experimental periodontitis group (EP) as well as in experimental periodontitis + ethanol (EP + EtOH) a decrease in Tb.Th parameter was observed compared to control and ethanol groups (p<0.0001). However, they were similar indicating that ethanol exposure per se promoted Tb.Th reduction and it was not intensified by experimental periodontitis induction.

In addition to Tb.Th, trabecular number parameter (Tb.N) was also evaluated. Similarly to Tb.Th data, Tb.N was reduced in Ethanol group compared to control (p = 0.0041) and when experimental periodontitis was induced a decrease in bone quality was also detected (i.e.,

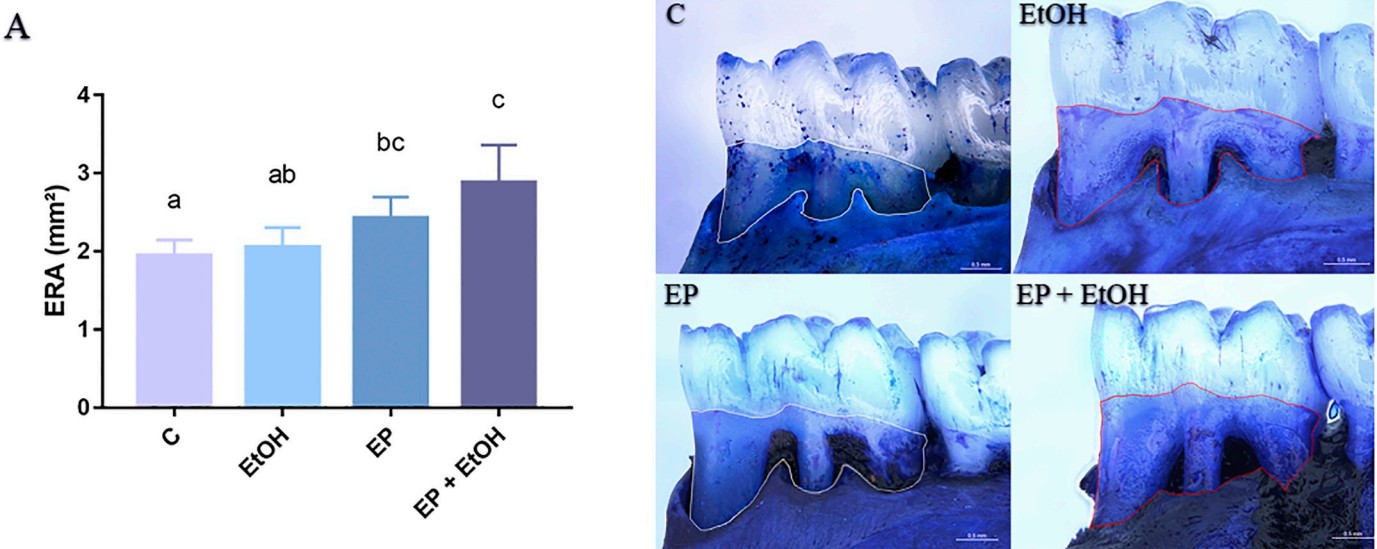

**Fig 2. Exposed root area in mm² in all experimental groups.** (A) Results are expressed as mean ± standard error of the number of exposed root area. On the right, representative photomicrographs of hemi-mandibles of the (C) Control group; (EtOH) Ethanol binge drinking group; (EP) Experimental periodontitis group and (EP + EtOH) Experimental periodontitis and Ethanol group. EP+EtOH group presented the highest ERA identified among the groups. Same letters indicate no significant differences between groups (P < 0.05, ANOVA followed by Tukey *post hoc* test). Scale bar: 0.5mm.

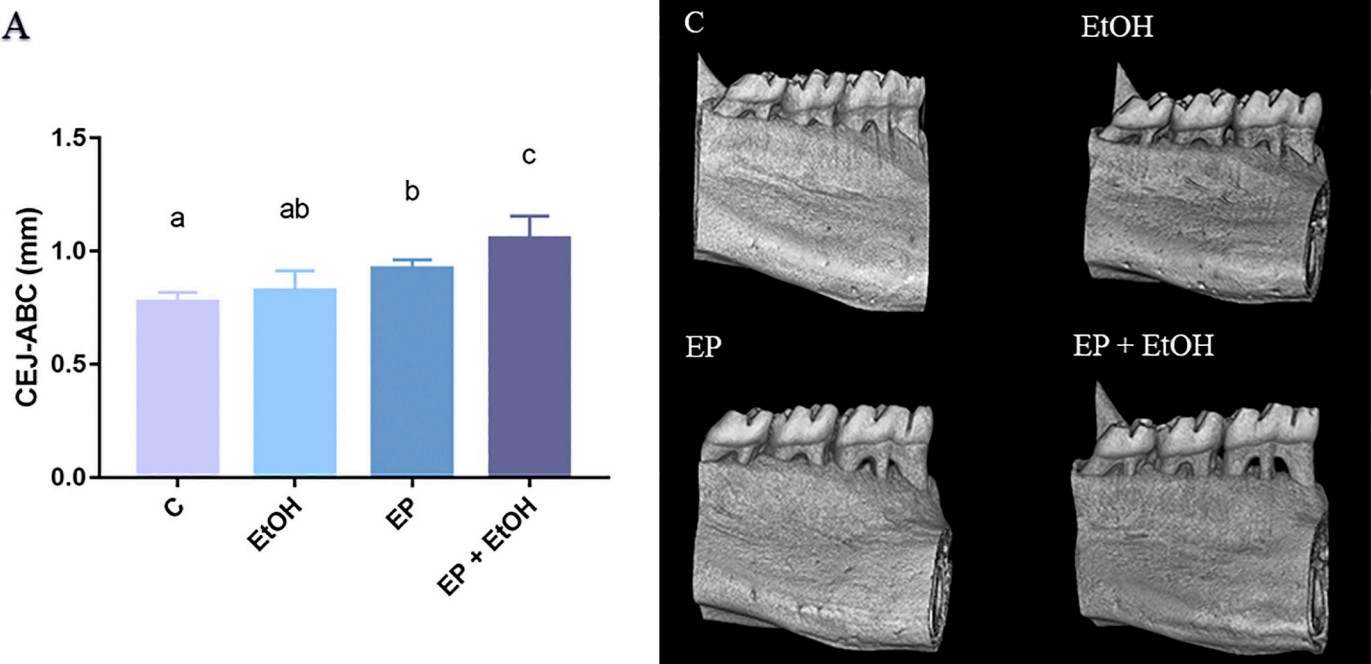

**Fig 3. Cementum-enamel junction to alveolar bone crest (CEJ-ABC) distance (mm) in all experimental groups.** (A) Results are expressed as mean ± standard error of the distance of cementum-enamel junction to alveolar bone crest in mm. On the right panel, there is the 3D reconstruction of hemi-mandibles of the (C) Control group; (EtOH) Ethanol binge drinking group; (EP) Experimental periodontitis group and (EP + EtOH) Experimental periodontitis and Ethanol group. EP +EtOH group presented the highest alveolar bone loss identified by the greatest measure of the ACJ-ABC distance. Same letters indicate no significant differences between groups (P < 0.05, ANOVA followed by Tukey *post hoc* test).

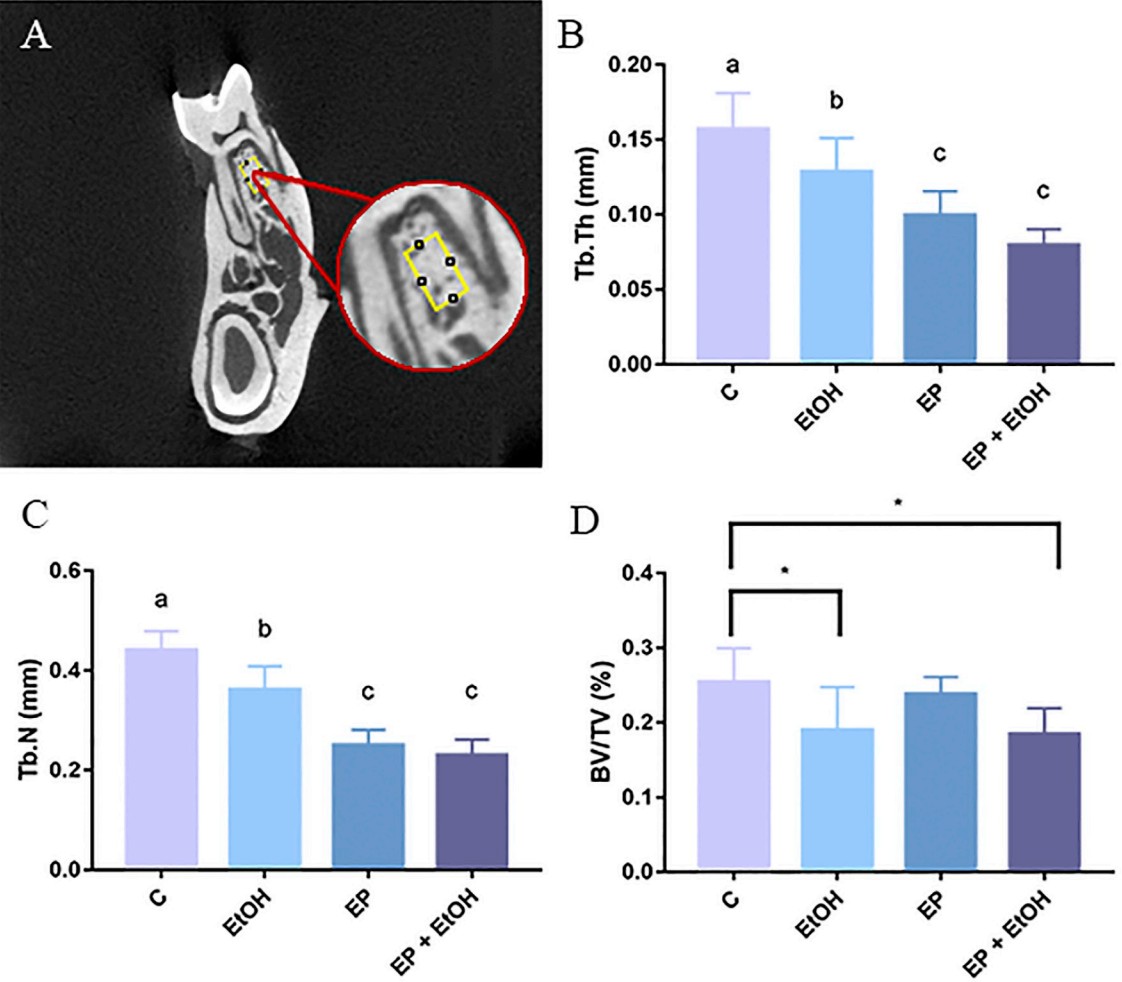

**Fig 4. Results from the Micro-CT quality analysis.** (A) Region of interest (ROI) used for evaluation. (B) Trabecular thickness parameter (Tb.Th) in all experimental groups. (C) Trabecular number parameter (Tb.N). For these results, same letters indicate no significant differences between groups (P < 0.05, ANOVA followed by Tukey post hoc test). (D) Bone density (%) parameter (BV/TV) obtained by micro-computed tomography (micro-CT). (*) indicates significant difference between the groups (p < 0.05).

ligature-induced periodontitis and ligature-induced periodontitis plus ethanol binge drinking groups; p<0.0001). In fact, no differences were observed between experimental periodontitis *versus* experimental periodontitis plus ethanol exposure groups (p = 0.8188), which is in accordance with Tb.Th evaluation, that reflects no synergistic pathological effects in this parameter.

Percent bone density (%BV/TV) was reduced solely in the ethanol binge drinking animals when compared to control group (p = 0.0411). Experimental periodontitis *per se* did not affect bone density. The graphic representation of all quality and density parameters are shown on Fig 4.

## Discussion

The present study aimed to investigate the effects of intermittent and episodic ethanol exposure in mandibular alveolar bone accompanied or not by experimental-induced periodontitis in rats. Our findings showed, for the first time, that intermittent and episodic ethanol exposure *per se* did not increase alveolar bone loss, however, the bone quality was altered in all

parameters evaluated. In addition, experimentally induced periodontitis aggravated the alveolar bone loss and trabecular thickness when ethanol intake was associated.

Different protocols have tested the effect of ethanol exposure on periodontal diseases in animal models with distinct results [6, 7, 10, 13, 22, 23]. The idea behind such studies comes from the demonstrated association between alcohol consumption, especially in high doses, with periodontal diseases in populational studies even in a longitudinal approach [6]. The studies performed in animals help in the understanding of the biological plausibility of the encountered associations in populations. Our group has investigated the effects of ethanol exposure on the stomatognathic system, both in chronic as well as in a binge drinking protocol [9–11]. In alveolar bone, our previous data demonstrated that chronic heavy ethanol exposure (6.5g/kg/day) from adolescence till adulthood led to increased alveolar bone loss, even in the absence of experimentally-induced periodontitis [10]. Contradictorily, other study reported that different ethanol concentrations consumed for 8 weeks did not increase alveolar bone loss, however, in the presence of ligature-induced periodontitis, alveolar bone loss was higher than in the control group [23].

In the present investigation, we verified that intense and sporadic alcohol intake promoted changes in quality and density of alveolar bone alone and in association with ligature-induced periodontitis. The 3 days On- 4 days-Off ethanol (3g/kg/day) protocol has been tested in animal model, the results demonstrated that it is a valid way to simulate binge drinking [24–26]. Binge drinking protocol has been widely used for investigation related to bone tissue alterations provoked by alcohol intake in a rat model. It has been described that alcohol exposure displayed a disordered expression of genes that play a pivotal role in bone remodeling [27]. In another study, bone resorption under orthodontic treatment was intensified by binge pattern exposure [28]. In addition, reduced bone mineral density and changes in the number of bone tissue biomarkers, such as receptor activator of nuclear factor-kappa-B ligand (RANKL) and osteoprotegerin (OPG), has been attributed to ethanol intake [29]. However, it is not well established the effects of binge drinking on the vertical alveolar bone loss, as well as on the alveolar bone quality in adult rats. Besides, we investigated if ethanol exposure interferes on alveolar bone loss and quality displayed by ligature-induced periodontal disease.

In order to verify the effects of binge drinking on the vertical alveolar bone loss, as well as on the alveolar bone quality micro-CT was employed [30–33]. Micro-CT has become the gold standard for bone tissue morphology and microstructure analysis in small animal models, such as rats and mice, as it enables high-definition three-dimensional reconstruction of samples and evaluate both cortical and trabecular bone [34]. Firstly, our data revealed that 3 days On- 4 days Off ethanol exposure protocol did not increase vertical alveolar bone loss. The previous study of our group demonstrated that daily administered heavy ethanol exposure from adolescence till adulthood increased alveolar bone loss [10]. These findings suggest that the harmful effects of ethanol exposure on alveolar bone loss, as well as other body tissues, depends on the period of life, as well as the duration of exposure and amount of consumption. In addition to binge drinking exposure, on the 14th day of ethanol exposure, we submitted animals from two groups to ligature-induced periodontitis. Animals that received the associated protocol (ligature-induced periodontitis + binge drinking) presented higher alveolar bone loss compared to the others that were submitted to experimental periodontitis and ethanol exposure protocols. These data suggest that prolonged ethanol exposure may aggravate alveolar bone loss on the presence of periodontal disorder. These results could be explained by the fact that alcohol exposure promotes oxidative damage and pro-inflammatory effects affecting bone homeostasis, accelerating bone resorption process as well as inhibiting bone formation [35–37]. Additionally, recent studies have shown that bone density is reduced by heavy ethanol consumption, with osteocalcin decrease and RANK / OPG ratio increase [8, 22, 35].

Although increased alveolar bone loss was not observed in the group that was solely exposed to ethanol, the alveolar bone quality was negatively altered by binge drinking. In fact, trabecular thickness, trabecular number, and percent bone density was affected by ethanol-exposed animals. A recent study reported that trabecular structures (i.e., thickness and number) were reduced in sub-chronic ethanol consumption protocol [38], which supports our findings. Likewise, it is well established that ethanol consumption displays oxidative unbalance and pro-inflammatory processes [39].

Another very interesting finding of our study is the fact that that ethanol exposure associated with experimental periodontitis aggravated alveolar bone loss and also disrupted bone quality indicators. Similar to ethanol consumption, periodontitis is also related to oxidative imbalance [40, 41]. We suggest that the pathological mechanisms shared by both isolated harmful stimuli, such as ethanol exposure and experimental periodontal disease, may provoke negative synergistic effects, intensifying alveolar bone tissue damage. Alcohol has been studied as a modifying factor of periodontal disease. In humans, a systematic review shows that several studies demonstrated an association between ethanol consumption and dependence with periodontitis. Although the heterogeneity between the articles is high, differing in the method of sample selection, alcohol consumption and alcohol use, periodontal indices and statistical analysis, the trend clearly demonstrates a detrimental effect [42]. Regarding the limits in translational researches, our results using pre-clinical animal model corroborate the idea that alcohol could negatively impact the pathogenesis of periodontitis. One possible explanation for that, is that the increase of C-reactive protein levels, as well as decrease of neutrophil chemotaxis and phagocytosis elicited by ethanol exposure may predispose to infection by periodontal bacteria [6]. In addition, TNF-$\alpha$ and IL-6 levels are higher in alcohol users [29], which might be equally in crevicular fluid, being related to the presence of periodontitis [6]. Furthermore, it is noteworthy that alcohol modulates bone cell activities in a dose-dependent manner. Thus, ethanol exposure causes an imbalance of the OPG / RANK / RANKL system, implicating in reduced osteoblast activity and osteoclastogenesis improvement [8, 10, 36], which consists of processes that also occurs during periodontitis development [43].

## Conclusion

It may be concluded that the intense and episodic ethanol intake ("binge pattern") promoted decrease in quality of alveolar bone in all parameters analyzed. In addition, we demonstrated that experimentally-induced periodontitis was aggravated binge drinking-like alcohol exposure, qualitatively and quantitatively. Thus, alcohol exposure is possibly related to the modification of the progression periodontitis, intensifying bone tissue damage.

## Supporting information

**S1 File.**
(DOCX)

## Acknowledgments

The authors are grateful to the Brazilian National Council for Scientific and Technological Development (CNPq) and Programa Nacional de Cooperação Acadêmica na Amazônia—PROCAD/Amazônia da Coordenação de Aperfeiçoamento de Pessoal de Nível Superior (CAPES).

## Author Contributions

**Conceptualization:** Deborah Ribeiro Frazão, Railson de Oliveira Ferreira, Rafael Rodrigues Lima.

**Formal analysis:** Deborah Ribeiro Frazão, Manoela Domingues Martins, Rafael Rodrigues Lima.

**Funding acquisition:** Rafael Rodrigues Lima.

**Investigation:** Deborah Ribeiro Frazão, Victória dos Santos Chemelo, Deiweson Monteiro, Railson de Oliveira Ferreira, Leonardo Oliveira Bittencourt, Gabriela de Souza Balbinot, Rafael Rodrigues Lima.

**Methodology:** Deborah Ribeiro Frazão, Deiweson Monteiro, Railson de Oliveira Ferreira, Leonardo Oliveira Bittencourt, Gabriela de Souza Balbinot, Rafael Rodrigues Lima.

**Project administration:** Deborah Ribeiro Frazão.

**Supervision:** Victória dos Santos Chemelo, Fabrício Mezzomo Collares, Cassiano Kuchenbecker Rösing, Rafael Rodrigues Lima.

**Validation:** Cristiane do Socorro Ferraz Maia, Fabrício Mezzomo Collares, Cassiano Kuchenbecker Rösing, Rafael Rodrigues Lima.

**Writing – original draft:** Deborah Ribeiro Frazão, Cristiane do Socorro Ferraz Maia, Manoela Domingues Martins, Rafael Rodrigues Lima.

**Writing – review & editing:** Cristiane do Socorro Ferraz Maia, Fabrício Mezzomo Collares, Cassiano Kuchenbecker Rösing, Manoela Domingues Martins, Rafael Rodrigues Lima.

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
