## [Decision Letter · Decision Letter 0]

21 May 2020

PONE-D-20-02327

Ethanol binge drinking exposure affects alveolar bone quality and aggravates bone loss in experimentally-induced periodontitis

PLOS ONE

Dear Dr Lima,

Thank you for submitting your manuscript to PLOS ONE. After careful consideration, we feel that it has merit but does not fully meet PLOS ONE’s publication criteria as it currently stands. Therefore, we invite you to submit a revised version of the manuscript that addresses the points raised during the review process.

We would appreciate receiving your revised manuscript by Jul 05 2020 11:59PM. To enhance the reproducibility of your results, we recommend that if applicable you deposit your laboratory protocols in protocols.io, where a protocol can be assigned its own identifier (DOI) such that it can be cited independently in the future. For instructions see: http://journals.plos.org/plosone/s/submission-guidelines#loc-laboratory-protocols

We look forward to receiving your revised manuscript.

Kind regards,

Timothy Damron

Academic Editor

PLOS ONE

Journal Requirements:

2. To comply with PLOS ONE submissions requirements, please provide methods of sacrifice in the Methods section of your manuscript

"The authors are grateful to the Brazilian National Council for Scientific and Technological Development (CNPq), Programa Nacional de Cooperação Acadêmica na Amazônia—PROCAD/Amazônia da Coordenação de Aperfeiçoamento de Pessoal de Nível Superior (CAPES), Brazilian Federal  Agency for Support and Evaluation of Graduate Education (CAPES- finance code 001), Pró-Reitoria de Pesquisa e Pós- Graduação PROPESP-UFPA."

"The funders had no role in study design, data collection and analysis, decision to

publish, or preparation of the manuscript."

5. Please ensure that you refer to Figure 2 in your text as, if accepted, production will need this reference to link the reader to the figure.

Reviewers' comments:

Reviewer's Responses to Questions

**Comments to the Author**

1. Is the manuscript technically sound, and do the data support the conclusions?

Reviewer #1: No

Reviewer #2: Yes

2. Has the statistical analysis been performed appropriately and rigorously? 

Reviewer #1: No

Reviewer #2: Yes

3. Have the authors made all data underlying the findings in their manuscript fully available?

Reviewer #1: Yes

Reviewer #2: Yes

4. Is the manuscript presented in an intelligible fashion and written in standard English?

Reviewer #1: Yes

Reviewer #2: Yes

5. Review Comments to the Author

Reviewer #1: The manuscript by Frazão and colleagues explores the potential role of ethanol consumption on alveolar bone loss. The study focused on the use of a rats to investigate whether ethanol exposure per se was sufficient to induce dental changes. It is an interesting and important question, and the group used a standard, well-accepted method for inducing periodontal tissue breakdown by placing ligatures around the molars. Four experimental groups were described: a “control” group, an ethanol exposure group in which rats are given ethanol for 3 days then off for 4, for a total of 28 days; another group that received ethanol and ligatures, and a fourth group that has ligatures and ethanol exposure as well.

If I understand the experimental groups correctly, those in G1 are not exposed to ethanol (the control group). Animals in G2 are exposed to ethanol (“Ethanol group (EtOH, G2) received ethanol binge drinking protocol characterized by 30% w/v ingestion…”). Animals in the G3 and G4 groups were also exposed to ethanol for 14 days (“On the 14th day of ethanol exposure protocol, animals from experimental periodontitis (G3) group and experimental periodontitis + ethanol (G4) group were submitted to… ligature-induced periodontal disease by insertion of cotton ligatures”). Consequently, there is no group that received ligatures to induce periodontitis) but did not receive ethanol. Therein lies my major concern with this work: without this group, it is impossible to separate the effects of the ligatures from the effects of ethanol exposure. Consequently, the data do not support the conclusion made by the authors, that binge-drinking decreases alveolar bone quality. (I would prefer that a measurable value be substituted since “bone quality” is a vague and non-quantitative descriptor). Other questions and concerns are outlined below, but in its current form the data do not support the conclusions drawn.

Animals in G4, which had both received ligatures and were exposed to ethanol presented a higher area of exposed root in comparation to “ethanol per se protocol and controls”. I don’t understand this result. If I go by the figure, the dark purple bar representing the G4 group is higher than the light purple G1 bar e.g., there is greater root exposure in G4 compared to G1 but that is easily understood by the fact that there is a ligature in the G4 group; it is expected that the ligature causes gingival recession.

The results in Figure 3 are described as follows: “animals submitted to binge drinking protocol plus experimental periodontitis presented an increase in alveolar bone loss compared to ethanol per se protocol.” The appropriate control, however, is the group that had ligatures but no ethanol exposure; this group does not exist, as far as I can tell.

In Figure 4, it appears that G3 and G4 exhibit an equivalent e.g., non-significant difference in BV/TV, trabecular number, and trabecular thickness. If I understand the groups correctly, that means that whether the animal is exposed to ethanol or not, the effect on alveolar bone loss is equivalent. Thus, I cannot see how

Reviewer #2: Dear Author, congratulations on a well designed study. It is well written and well performed. However, I am missing some more parameters from this study. Since you already did microCT, I think it would be benefical for this study to measure the distance between the cemento-enamel junction and the alveolar bone crest (CEJ-AC) at 6 different points (mesio-buccal, buccal, distobuccal, disto-lingual, lingual and mesio-lingual), noy just the four points that you have chosen, particular when you see the large different in alveolar bone promixal and distal. Please see Virto L, , et al. Melatonin expression in periodontitis and obesity: An experimental in-vivo investigation. J Periodont Res. 2018;53:825–831. on how to do it. It does not take long time, and would give you valuable information on the bone loss similar to how you would evaluated with a periodontal probe in the clinic. It would be beneficial for this study to include histology

6. PLOS authors have the option to publish the peer review history of their article (what does this mean?). If published, this will include your full peer review and any attached files.

Reviewer #1: No

Reviewer #2: No

---

## [Author Response · Author response to Decision Letter 0]

3 Jun 2020

Dear Reviewers,

Thank you for your dedication to revise our manuscript. In this document are the answers to the comments made by reviewers 1 and 2. The answers are described in detail and everything that has been changed in the manuscript is described here. 

Referee: 1

1. “[…] Four experimental groups were described: a “control” group, an ethanol exposure group in which rats are given ethanol for 3 days then off for 4, for a total of 28 days; another group that received ethanol and ligatures, and a fourth group that has ligatures and ethanol exposure as well. If I understand the experimental groups correctly, those in G1 are not exposed to ethanol (the control group). Animals in G2 are exposed to ethanol (“Ethanol group (EtOH, G2) received ethanol binge drinking protocol characterized by 30% w/v ingestion…”). Animals in the G3 and G4 groups were also exposed to ethanol for 14 days (“On the 14th day of ethanol exposure protocol, animals from experimental periodontitis (G3) group and experimental periodontitis + ethanol (G4) group were submitted to… ligature-induced periodontal disease by insertion of cotton ligatures”). Consequently, there is no group that received ligatures to induce periodontitis) but did not receive ethanol […]”

Answer: 

We apologize for the misunderstanding about that information. We noticed that it was not very clear in the text that the Experimental Periodontitis (EP, G3) group of this work was only submitted to ligature, but without receiving ethanol, receiving distilled water instead, as described in Figure 1. We have already modified this information in the manuscript to increase understanding. Group 2 and group 3 were submitted to separate exposures: one just received ethanol (G2) and the other received induction of experimental periodontitis, without ethanol exposure (G3). Thus, G2 and G3 represent the groups that were exposed to the challenges separately, while Group 4 represents the group that received both challenges (ethanol and ligature-experimental periodontitis) simultaneously.

2. “Animals in G4, which had both received ligatures and were exposed to ethanol presented a higher area of exposed root in comparation to “ethanol per se protocol and controls”. I don’t understand this result. If I go by the figure, the dark purple bar representing the G4 group is higher than the light purple G1 bar e.g., there is greater root exposure in G4 compared to G1 but that is easily understood by the fact that there is a ligature in the G4 group; it is expected that the ligature causes gingival recession.”

Answer: 

We understand your question and the opportunity to clarify it. When the G4 (EP + EtOH) is compared with G2 (EtOH), we observe a statistically significant difference, that leads us to understand that ethanol per se did not cause an increase in the exposed root area, but when associated with experimental periodontitis, i.e, when there is a synergistic effect of the two exposures, demonstrated by the increased area. This analysis by stereomicroscope is complementary to the evaluation of alveolar bone loss in height by micro-CT, since it shows us not only how much bone was lost in height, but the root area left exposed due to the process of periodontal breakdown. 

3. “The results in Figure 3 are described as follows: “animals submitted to binge drinking protocol plus experimental periodontitis presented an increase in alveolar bone loss compared to ethanol per se protocol.” The appropriate control, however, is the group that had ligatures but no ethanol exposure; this group does not exist, as far as I can tell.”

Answer: 

In fact, this group exists, it is group 3, which only underwent to ligature-induced periodontitis without exposure to ethanol. As shown in the graph, in figure 3, and in the text, on lines 137-140, there was a significant difference between group 3 and group 4, showing that when associating periodontitis with ethanol, there was a greater bone loss in height compared to only experimental periodontitis induction.

4. “In Figure 4, it appears that G3 and G4 exhibit an equivalent e.g., non-significant difference in BV/TV, trabecular number, and trabecular thickness. If I understand the groups correctly, that means that whether the animal is exposed to ethanol or not, the effect on alveolar bone loss is equivalent. Thus, I cannot see how.”

Answer: 

We understand your question. The data shows that the EtOH (G2) group presented a greater loss of trabecular thickness, trabecular number, and BV/TV compared to the control group. However, the groups with experimental periodontitis (G3 and G4) had a much greater loss compared to G1 and G2, but that was not significantly different from each other. This leads us to believe that because of periodontitis, the microstructural parameters were much more affected compared to the damage caused only by ethanol, which can be explained by the increased inflammation in groups with periodontitis.

Referee: 2

1. “Dear Author, congratulations on a well designed study. It is well written and well performed. However, I am missing some more parameters from this study. Since you already did microCT, I think it would be benefical for this study to measure the distance between the cemento-enamel junction and the alveolar bone crest (CEJ-AC) at 6 different points (mesio-buccal, buccal, distobuccal, disto-lingual, lingual and mesio-lingual), noy just the four points that you have chosen, particular when you see the large different in alveolar bone promixal and distal. Please see Virto L, , et al. Melatonin expression in periodontitis and obesity: An experimental in-vivo investigation. J Periodont Res. 2018;53:825–831. on how to do it. It does not take long time, and would give you valuable information on the bone loss similar to how you would evaluated with a periodontal probe in the clinic. It would be beneficial for this study to include histology”

Answer:

Thank you for your contribution and evaluation of the manuscript. We already evaluated the alveolar bone loss in the micro-CT at 6 points, as described in the lines 95-98 of the article. However, to facilitate understanding, the name of the points used for the evaluation was modified according to the suggested reference (mesio-buccal, buccal, distobuccal, disto-lingual, lingual, and mesio-lingual). 

Regarding the histological evaluation, for this work (the first with binge drinking paradigm) we tried to make a diagnostic analysis of this additional exposure pattern. We appreciate the suggestion and intend to use it for future work to evaluate the mechanisms of damage to alveolar bone associated with ethanol binge drinking consumption and experimental periodontitis. 

Finally, we would like to thank you for the meaningful contributions and acknowledge that they have contributed substantially to improve the impact of our manuscript. We hope that the changes implemented and the answers provided are enough for the recommendation of the publication of our study in Plos One.

---

## [Decision Letter · Decision Letter 1]

1 Jul 2020

Ethanol binge drinking exposure affects alveolar bone quality and aggravates bone loss in experimentally-induced periodontitis

PONE-D-20-02327R1

Dear Dr. Lima,

We’re pleased to inform you that your manuscript has been judged scientifically suitable for publication and will be formally accepted for publication once it meets all outstanding technical requirements.

Kind regards,

Timothy Damron

Academic Editor

PLOS ONE

Additional Editor Comments (optional):

Reviewers' comments:

Reviewer's Responses to Questions

**Comments to the Author**

1. If the authors have adequately addressed your comments raised in a previous round of review and you feel that this manuscript is now acceptable for publication, you may indicate that here to bypass the “Comments to the Author” section, enter your conflict of interest statement in the “Confidential to Editor” section, and submit your "Accept" recommendation.

Reviewer #2: All comments have been addressed

2. Is the manuscript technically sound, and do the data support the conclusions?

Reviewer #2: Yes

3. Has the statistical analysis been performed appropriately and rigorously? 

Reviewer #2: Yes

4. Have the authors made all data underlying the findings in their manuscript fully available?

Reviewer #2: Yes

5. Is the manuscript presented in an intelligible fashion and written in standard English?

Reviewer #2: Yes

6. Review Comments to the Author

Reviewer #2: no more comments need for this round and I hereby accept the manuscript for publication. Thank you for address the comments made

7. PLOS authors have the option to publish the peer review history of their article (what does this mean?). If published, this will include your full peer review and any attached files.

Reviewer #2: No

---

## [Editor Report · Acceptance letter]

20 Jul 2020

PONE-D-20-02327R1 

Ethanol binge drinking exposure affects alveolar bone quality and aggravates bone loss in experimentally-induced periodontitis 

Dear Dr. Lima:

I'm pleased to inform you that your manuscript has been deemed suitable for publication in PLOS ONE. Congratulations! Your manuscript is now with our production department. 

Kind regards, 

on behalf of

Dr. Timothy Damron 

Academic Editor

PLOS ONE